# The Role of Plasma Extracellular Vesicles in Remote Ischemic Conditioning and Exercise-Induced Ischemic Tolerance

**DOI:** 10.3390/ijms23063334

**Published:** 2022-03-19

**Authors:** Tingting Gu, Jesper Just, Katrine Tang Stenz, Yan Yan, Peter Sieljacks, Jakob Wang, Thomas Skjaerlund Groennebaek, Jesper Emil Jakobsgaard, Emil Rindom, Jon Herskind, Anders Gravholt, Thomas Ravn Lassen, Mathias Jørgensen, Rikke Bæk, Eugenio Gutiérrez-Jiménez, Nina Kerting Iversen, Peter Mondrup Rasmussen, Jens Randel Nyengaard, Malene Møller Jørgensen, Frank de Paoli, Hans Erik Bøtker, Jørgen Kjems, Kristian Vissing, Kim Ryun Drasbek

**Affiliations:** 1Center of Functionally Integrative Neuroscience, Department of Clinical Medicine, Aarhus University, 8000 Aarhus, Denmark; gutt@cfin.au.dk (T.G.); jesperj@clin.au.dk (J.J.); kts@cfin.au.dk (K.T.S.); eugenio@cfin.au.dk (E.G.-J.); nkiversen@cfin.au.dk (N.K.I.); pmr@cfin.au.dk (P.M.R.); 2Sino-Danish College (SDC), University of Chinese Academy of Sciences, Beijing 100190, China; jrnyengaard@clin.au.dk; 3Department of Molecular Medicine, Aarhus University Hospital, 8200 Aarhus, Denmark; 4Interdisciplinary Nanoscience Center, Aarhus University, 8000 Aarhus, Denmark; yanyan@inano.au.dk (Y.Y.); jk@mbg.au.dk (J.K.); 5Exercise Biology, Department of Public Health, Aarhus University, 8000 Aarhus, Denmark; petersieljacks@gmail.com (P.S.); jawa@ph.au.dk (J.W.); tsg@nmdpharma.com (T.S.G.); jj@ph.au.dk (J.E.J.); rindom@bio.au.dk (E.R.); jherskind@ph.au.dk (J.H.); a.gravholt@live.dk (A.G.); 6Department of Cardiology, Aarhus University Hospital, 8200 Aarhus, Denmark; thomasravnl@clin.au.dk (T.R.L.); boetker@au.dk (H.E.B.); 7Department of Engineering, Aarhus University, 8000 Aarhus, Denmark; mathias.lindh.j@gmail.com; 8Department of Clinical Immunology, Aalborg University Hospital, 9100 Aalborg, Denmark; rikke.baek@rn.dk (R.B.); maljoe@rn.dk (M.M.J.); 9Core Center for Molecular Morphology, Section for Stereology and Microscopy, Department of Clinical Medicine, Aarhus University, 8000 Aarhus, Denmark; 10Department of Pathology, Aarhus University Hospital, 8200 Aarhus, Denmark; 11Department of Clinical Medicine, Aalborg University, 9220 Aalborg, Denmark; 12Department of Biomedicine–Physiology, Aarhus University, 8000 Aarhus, Denmark; fdp@biomed.au.dk; 13Department of Molecular Biology and Genetics, Aarhus University, 8000 Aarhus, Denmark

**Keywords:** remote ischemic conditioning, resistance exercise, brain ischemia, stroke, plasma extracellular vesicles, human brain vascular endothelial cells, cell viability, angiogenesis, infarction

## Abstract

Ischemic conditioning and exercise have been suggested for protecting against brain ischemia-reperfusion injury. However, the endogenous protective mechanisms stimulated by these interventions remain unclear. Here, in a comprehensive translational study, we investigated the protective role of extracellular vesicles (EVs) released after remote ischemic conditioning (RIC), blood flow restricted resistance exercise (BFRRE), or high-load resistance exercise (HLRE). Blood samples were collected from human participants before and at serial time points after intervention. RIC and BFRRE plasma EVs released early after stimulation improved viability of endothelial cells subjected to oxygen-glucose deprivation. Furthermore, post-RIC EVs accumulated in the ischemic area of a stroke mouse model, and a mean decrease in infarct volume was observed for post-RIC EVs, although not reaching statistical significance. Thus, circulating EVs induced by RIC and BFRRE can mediate protection, but the in vivo and translational effects of conditioned EVs require further experimental verification.

## 1. Introduction

Ischemic stroke is caused by restriction of blood supply to an area causing a shortage of oxygen and nutrients, which is usually a consequence of blood clot formation in the artery or an embolus traveling from other parts of the body, e.g., the heart. The occlusion of blood flow leads to the failure of energy-consuming ion pumps and intracellular accumulation of Na^+^ and Ca^2+^. Overloading of intracellular Na^+^ and Ca^2+^ causes mitochondrial dysfunction and can trigger apoptosis or necrosis. Mitochondrial dysfunction also leads to an overproduction of reactive oxygen species (ROS) that induces oxidative stress. In all, this results in immediate cell death in the ischemic core [1,2,3], whereas surrounding areas with reduced perfusion, i.e., the penumbra, may maintain the metabolism of active cells and be salvageable upon timely reperfusion [4,5]. Although many potential experimental treatment targets have been revealed during the past years, none of them have been successfully translated into clinical practice, and stroke remains a leading cause of long-term disability worldwide [6]. One explanation for the low clinical success rate might be that previous approaches to alleviate ischemic damage predominantly targeted the above-mentioned mechanisms individually, thus disregarding the true physiological interplay between mechanisms involved in tissue damage after the occurrence of ischemic stroke.

Ischemic conditioning and exercise stimulation have been observed to mediate ischemic tolerance when applied either before or after an ischemic insult [7,8,9,10]. Ischemic conditioning under resting conditions consists of repeated transient ischemia-reperfusion in a localized area [11] or in a limb as remote ischemic conditioning (RIC) [12,13] that protects against longer and more detrimental ischemia. The working mechanism underlying RIC has been studied by multiple preclinical studies from different aspects, including the decrease of oxidative stress by initiating the antioxidant Nrf2-ARE pathway [14], regulation of inflammatory responses by blocking NFκB [15], and upregulation of autophagy through the IL-6-dependent JAK-STAT pathway [16,17]. In fact, transient alternations of ischemia and reperfusion are also inherent in muscle activity. Accordingly, during conventional high-load resistance exercise (HLRE), the muscle produces large transient increases in tissue pressure followed by relaxation, leading to brief ischemia-reperfusion periods during movement cycles. HLRE has been proven to contribute to functional recovery after ischemic stroke [18]. However, individuals experiencing ischemic incidents are often elderly individuals with comorbidities, which may prohibit high loading during training [19]. Interestingly, low-load resistance exercise with simultaneous partial blood flow occlusion (blood-flow-restricted resistance exercise, BFRRE) offers an alternative and feasible rehabilitation strategy, which because of accelerated fatigue [20] has been shown to enhance muscle accretion and mitochondrial properties in healthy individuals, as well as functional capacity in, for example, chronic heart-failure patients [21,22,23], while still requiring a very low work volume. Combining low load with blood-flow restriction in BFRRE will collectively magnify ischemic stimulation. Thus, exercise regimes, and especially BFRRE, may share the underlying mechanisms and protective capabilities of RIC. Therefore, revealing the underlying mechanism of how these different approaches trigger endogenous protection against brain ischemia and reperfusion may provide new perspectives for novel neuroprotective treatments.

Extracellular vesicles (EVs) have been suggested as potential mediators of the protective effects. ‘EVs’ is the collective term for small vesicles (diameter of 30 nm to 1000 nm) that include exosomes, microvesicles, ectosomes, apoptotic bodies, etc. They are unable to replicate and are encased by a lipid bilayer membrane that makes them stable in the extracellular space and body fluids [24,25,26]. EVs are known to mediate intercellular communications in both local and long-distance inter-organ environments by carrying information in the forms of proteins, lipids, and nucleic acids in physiological or pathological conditions [27]. EVs originate from a large array of cell types with the majority of EVs found in circulation coming from blood platelets, erythrocytes, and monocytes [28]. However, muscle-specific EVs have been described to swiftly increase during aerobic exercise and account for 1–5% of the total EV pool [29]. The concentration of circulating EVs have been shown to increase immediately after RIC and different modalities of aerobic and resistance exercise, however, discrepancies exist, as several studies have also found no changes in the concentration of plasma EVs after RIC or exercise [30,31]. In a previous study [32], we demonstrated how BFRRE-conditioned EVs could stimulate primary skeletal muscle proliferation. In addition, plasma EVs secreted after RIC have been shown to reduce infarct size after myocardial ischemia-reperfusion injury [33,34] and EVs can cross the blood-brain barrier (BBB), extending their possible range of impact to cells in the brain [35]. All these characteristics and evidence point to EVs as possible mediators of ischemic conditioning and exercise that increase cerebral ischemic tolerance.

Based on these recent findings, we hypothesized that transient sub-lethal ischemic conditioning conducted as RIC or resistance exercise regimens can mediate EV release in plasma that is able to convey remote brain protection against ischemia from stroke. To test this hypothesis in a comprehensive translational experimental and methodological setting, we investigated the protective role of EVs isolated from the plasma of human volunteers undergoing RIC, BFRRE, and HLRE, and from a non-intervention control (NIC) group. After EV characterization, including morphology, concentration, and size, the effects of the EVs were screened in different in vitro assays using endothelial cells as model cells. Endothelial cells are of special interest as they are a part of the BBB that line the inside of blood vessels and thus are the first cells to encounter bloodborne particles. Finally, considering the clinical translatability of RIC in acute treatment of ischemic stroke, the RIC EVs underwent testing in an in vivo ischemic stroke murine model to investigate their potential tissue protective effect in the acute phase (4 h reperfusion following occlusion) and on functional outcome during the first 7 days after an experimental ischemic stroke (Figure 1).

## 2. Results

### 2.1. Fluctuations in Extracellular Vesicles after Single Bout Intervention

To investigate changes in the circulating EV profile, blood was collected before and at three timepoints after NIC, RIC, BFRRE, and HLRE (pre-, 5 min post-, 30 min post-, and 6 weeks post-intervention). EVs were isolated using size exclusion chromatography and characterized by nanoparticle tracking analysis (NTA), transmission electron microscopy (TEM), and Western blotting (WB). The concentration of circulating NIC, RIC, BFRRE, and HLRE EVs did not change at any timepoint after intervention. Only minor, non-significant fluctuations were observed (Figure 2A). Likewise, no change in the mean or peak size of the vesicles was observed (Figure 2B,C). EV morphology was visualized by TEM in ultrathin sections of epon-embedded EVs (Figure 2D), and the presence of the EV marker Flotillin 1 was verified by Western blotting (Figure 2E).

Next, we analyzed 44 pre-selected EV surface markers using the EV array. We were only able to robustly measure the expression of eight markers (CD3, CD9, CD16, CD25, CD31, CD81, LRP-1, and ICAM-1), including the canonical EV markers CD9 and CD81 (Appendix A). The robust presence of these surface markers pointed toward blood cells (CD3 (T-cell), CD16 (NK cells, neutrophils, monocytes and macrophages), CD25 (B lymphocytes), and endothelial cells (CD31 and ICAM-1) as the origin of a major fraction of the circulating EVs. No significant change in the surface marker profile of these eight markers was observed for the NIC, RIC, or HLRE intervention groups. However, for the BFRRE group, a small up-regulation of CD81 was observed at 5 and 30 min post-intervention.

### 2.2. EVs Released after RIC and BFRRE Protect Endothelial Cells against Prolonged Oxygen and Glucose Deprivation

As endothelial cells are the first to encounter circulating EVs, we tested whether cultured human brain microvascular endothelial cells (HBMECs) could take up EVs. Indeed, human post-RIC EVs were found inside HBMECs, indicating that endothelial cells can take up EVs from the blood (Figure 3A,B). To test the protective potential of conditioned EVs, we subjected HBMECs to oxygen-glucose deprivation (OGD) in vitro that mimics ischemia and reperfusion and tested cell viability by ATP production. Post-RIC EVs (5 min), post-BFRRE EVs (5 min), and post-HLRE (5 min and 30 min) exhibited protective effects (*p* = 0.016, *p* = 0.019, *p* = 0.030, and *p* = 0.020, respectively), while EVs from the remaining time points, as well as EVs from the NIC group, had no effect (Figure 3C). However, only 5 min post-RIC EVs and 5 min post-BFRRE EVs also increased viability significantly compared to the EVs from the NIC group (*p* = 0.016 and *p* = 0.026) (Figure 3C).

### 2.3. EVs Released after RIC, BFRRE, and HLRE Do Not Affect Endothelial Cell Tube Formation In Vitro

Restoring oxygen delivery to the ischemic area is paramount for the survival of neurons when a stroke occurs. Intuitively, formation of new blood vessels would be important to restore delivery of oxygen to the ischemic area. Therefore, we investigated the angiogenic potential of conditioned EVs. Total branching length of EV-treated HBMECs was used as a proxy for tube formation and was measured at 2, 4, 6, and 24 h post seeding on Matrigel. As no effects of the 6-week post-intervention EVs were observed in the viability assay above, we only tested the 5- and 30-min post-intervention EVs (Figure 3D–I). For the positive control, the total branching length reached a maximum at 4–6 h while a reduction was observed at 24 h because of the assembly of the branches into fewer sturdier structures (Figure 3H). However, for some of the EV-treated cells, the total branching length remained high or increased from 6–24 h. (Appendix A). No significant changes between pre- and post-intervention EVs were seen at 2, 4, 6, or 24 h (Figure 3I). Furthermore, no significant changes were observed between the different intervention groups.

### 2.4. Pooled Post-RIC EVs Increase Endothelial Cell Survival In Vitro and Accumulate in the Ischemic Area of tMCAO Mice

To validate the protective effect of post-RIC EVs on HBMECs, we investigated whether EVs isolated 5 min after RIC intervention, in addition to cell viability, also influenced cell death. Furthermore, we used this cell death assay to confirm that pooled EVs from the different volunteers, within each group and time point, still carried a protective effect. Pooling the EVs was necessary to prepare sufficient EV aliquots for the in vivo experiments presented below and to decrease variability and minimize the number of animals used. As the RIC intervention is the most clinically relevant for acute ischemic stroke patients, for whom exercise is difficult to perform, we chose to only include RIC EVs in the acute transient MCAO stroke model. As above, HBMECs were subjected to OGD with the addition of pooled pre-RIC or post-RIC EVs (5 min post intervention). After OGD, cell death was determined by flow cytometry as the percentage of propidium iodide positive cells (Figure 3N). Post-RIC EVs decreased cell death compared to pre-RIC EVs and control (*p* = 0.002 and *p* < 0.001 respectively). Interestingly, pre-RIC EVs also decreased cell death (*p* = 0.005), indicating that circulatory plasma EVs themselves protect against anoxic endothelial cell death. Visual inspection showed that the number of cells with a healthy morphology was visibly increased for post-RIC EV treated cells, but also to some extent for pre-RIC EVs (Figure 3K–M).

Based on the translational potential of RIC and the above results, only early time-point RIC EVs were tested in the very laborious meticulous stroke mouse model, tMCAO. Both pre- and post-RIC EVs could be seen in the brain of tMCAO mice after 4 h of reperfusion (Figure 4). Neglectable fluorescence was seen in the no-EV vehicle control (vehicle, *n* = 4, Figure 4A), showing that excess free dye is effectively removed during EV labelling. Labelled pre-RIC EVs (*n* = 3, Figure 4B–D) were visible in the brains while post-RIC EVs (*n* = 4, Figure 4E–H) showed a clearer accumulation in the ischemic hemisphere compared to vehicle (Figure 4I) (*p* = 0.04). Furthermore, in three out of four animals, post-RIC EVs accumulated in the ischemic hemisphere (right-side hemispheres in Figure 4E–H,J) compared to the healthy hemisphere (*p* = 0.0016). These tracking data suggest that post-RIC EVs enter the brain region and specifically accumulate in the ischemic area.

### 2.5. Perfusion and Hemoglobin Changes after EV Administration in the tMCAO Stroke Model

Using LSCI together with MSRI, we studied perfusion, oxyHb, and deoxyHb changes during the first hours of reperfusion in pre-RIC (*n* = 8) and post-RIC EV (*n* = 7) treated mice, as well as vehicle-injected control mice (*n* = 7). Analysis masks were created to define the ischemic core (<33% of baseline), penumbra (33–70% of baseline), and normal perfused (>70% of baseline) brain tissue (Figure 5). An increase in perfusion in the ischemic core and penumbral area was observed immediately after reperfusion. However, no significant differences in perfusion changes between the treatment groups were seen in either the ischemic core or the penumbral area (Figure 5D–F). Besides, no significant group effects on oxyHb and deoxyHb changes were detected by mixed-model analysis, although EV-treated animals showed higher oxyHb concentration increases after reperfusion compared to the control animals throughout the reperfusion period, especially in the ischemic core and the penumbra (Figure 5G–I). As for the deoxyHb concentration changes, post-RIC EVs induced a larger decrease for the deoxyHb concentration in the first hour after reperfusion in the ischemic core, but not in the penumbral area (Figure 5J–L).

### 2.6. Post-RIC EVs Have No Impact on Neurological Function after Ischemic Stroke

Prolonging reperfusion to 7 days enabled us to measure improvements in neurological function following transient ischemic stroke. We found that mice from all groups got a mild to moderate level of focal neurological deficit based on neurological scoring on day 1 after surgery, with a gradual recovery from day 2 until day 7 (Figure 6A). Two-way ANOVA showed no significant differences (*p* = 0.96) between control (*n* = 11), pre-RIC (*n* = 12), and post-RIC EV (*n* = 11) treated mice. As expected, tMCAO in the left hemisphere, resulted in higher frequencies of turning right in all three groups for the corner test. However, no significant group differences were found by two-way ANOVA analysis (Figure 6B). For the Hargreave’s test, mice in all groups had a longer latency to withdrawal from the heating source in the right hind paw compared to the left hind paw 7 days after tMCAO surgery (Figure 6C), but no significant differences were observed between groups. These results suggest that neither pre-RIC EVs nor post-RIC EVs improved neurological function in tMCAO mice during the first 7 days after stroke.

### 2.7. Ischemic Core Development after EV Administration in tMCAO Mice

Laser speckle imaging of blood flow changes showed the development of the ischemic core during ischemia and the early stages of reperfusion. The ischemic core was stable during the study period (4 h reperfusion) for both pre-RIC (*n* = 8) and post-RIC EV (*n* = 7) treated animals (Figure 7A). In contrast, the ischemic core area in the control animals (*n* = 7) gradually increased over the last 30 min of the 4 h reperfusion period, resulting in a doubling of the ischemic core (increase from 6% to 12% of the hemisphere). The final infarct volumes at 4 h after reperfusion were estimated based on Map-2 staining (Figure 7B–D). No significant group effect on infarct size was detected by one-way ANOVA (*p* = 0.36), although compared with pre-RIC EVs (*n* = 8) and the control group (*n* = 6), animals receiving post-RIC Evs (*n* = 5) had a smaller mean of infarct volume (Figure 7H). Prolonged reperfusion (7d) resulted in larger infarcts that included both cortex and striatum for control (*n* = 10), pre-RIC *(n* = 7), and post-RIC EV (*n* = 8) treated animals (Figure 7E–G). However, no significant differences in the infarct size were observed between groups with one-way ANOVA (Figure 7I).

### 2.8. The Integrity of BBB at 4 h after Reperfusion

To detect the potential in vivo protection on endothelial cells by post-RIC EVs, we investigated BBB integrity 4 h after reperfusion by IgG extravasations in the infarct area, penumbra, and healthy hemisphere in the control (*n* = 6), pre-RIC (*n* = 8), and post-RIC EV (*n* = 5) treated mice (Appendix A). After normalizing IgG extravasation in the infarct and penumbra to the healthy area, we found a significantly higher level of IgG extravasation in the penumbral area compared to the healthy area (*p* = 0.04) when combining all groups, while no significant differences were found between groups in IgG extravasation in any of the areas. These data indicate that post-RIC EVs did not affect the integrity of the BBB at 4 h after reperfusion.

## 3. Discussion

In recent years, EVs have attracted immense attention based on their various physiological functions and pathological importance, as well as the potential use of EVs as a personalized drug delivery system [36,37,38]. In this regard, unravelling the role of EVs in conditions of ischemia-reperfusion, and whether EVs act as circulating mediators to provide endogenous protection against ischemic complications such as lethal cerebral ischemia, may provide valuable insight and perspectives for treatment.

To pursue this, we employed treatment protocols including ischemia-reperfusion +/− muscle contractile activity in healthy young human volunteers. Before and after single, as well as repeated treatments, plasma samples were collected and EVs were characterized and tested in the in vitro OGD assay and the angiogenesis assay on HBMECs. Finally, RIC EVs were tested in the tMCAO mouse stroke model during the acute phase (4 h reperfusion), as well as 7 days after ischemia.

Using a translational approach, our major findings comprise (1) that EVs released shortly after RIC and BFRRE possess the ability to increase endothelial cell viability and that RIC EVs reduce endothelial cell death after oxygen-glucose deprivation, and (2) the observation that labeled EVs injected during occlusion of tMCAO accumulate in the ischemic hemisphere acutely after ischemia, and (3) that post-RIC EVs show signs of promoting protection against infarction and functional decay early after transient ischemia in the tMCAO mouse model, although data could not robustly support this.

The human intervention protocols were designed to mimic commonly practiced RIC, BFRRE, and HLRE protocols [22,23,39,40]. Moreover, a separate NIC group was included, and sample collection was performed at identical time points for all volunteers. This improved the ability to distinguish genuine effects of intervention regimens from potentially confounding effects adhering to tissue collection, dietary premise, and diurnal rhythm, which have previously proven to be of principal importance to properly interpret acute responses in human experimental settings [41]. Finally, through direct measurements of plasma hemolysis, we carefully ascertained that blood samples used for this study exhibited low levels of hemolysis at all timepoints. This aspect is of particular importance as red blood cells contain EVs, which could be leaked during hemolysis [42,43]. Based on these collective precautions, which are regrettably not included in a multitude of previous human experimental investigations on systemic molecules in general and EVs specifically, we were able to discern more strongly between intervention effects and the aforementioned potentially confounding factors.

As for EV characterization, neither RIC nor the exercise interventions, BFRRE and HLRE, induced an increase in the concentration of circulating EVs at any timepoint after intervention, consistent with our earlier study of EVs following BFRRE [32]. However, this is not consistent with other reports, in which increased EV concentrations were observed in the circulation immediately after cessation of skeletal muscle activity [44,45,46,47]. Our findings support recent results demonstrating that four cycles of RIC did not cause an increase in EVs in patients or healthy controls [48]. Furthermore, diverging results on the impact of RIC on overall EV concentration have been reported [46,47], which may adhere to differences in standardization of RIC protocols, as well as EV purification and characterization methods (reviewed in [26]). Additionally, known confounders of EV concentration measurements include non-vesicular particles such as proteins and lipid complexes [49]. On the other hand, by using the same number of EVs from each condition in our in vitro assays, we demonstrated that the biological effect of EVs released immediately after RIC or BFRRE changed. Consequently, RIC and BFRRE most likely induced release of EVs with an altered composition of cargo and/or surface markers, and, as such, the circulating EV pool became more adept in preventing ischemic damage.

From the EV surface marker array analysis, we were able to compare the expression of 8 out of the 44 pre-selected markers. This was somewhat underwhelming compared to the utilization of a hypothesis-free approach like mass spectrometry where thousands of EV proteins can be detected [47]. Nonetheless, the expression of these eight markers unsurprisingly indicated that the circulating EVs originate from blood cells and endothelial cells [28,50]. Furthermore, the up-regulation of the canonical EV surface marker CD81 on EVs released immediately after BFRRE may point toward an acute release of EVs, albeit not one reflected in the EV concentration measured by NTA.

Through the in vitro assays, we assessed the influence of post-intervention plasma EVs on endothelial cells as these cells previously have been suggested as focal points of the protective effects of conditioning-based intervention strategies. A 5% plasma dilution mix isolated from participants subjected to RIC was previously shown to protect endothelial cells from subsequent ischemia reperfusion injury [51], while serum from RIC-treated rats has been shown to exert an antioxidant effect on endothelial cells [52]. In the present study, both post-RIC and post-BFRRE EVs isolated immediately after intervention improved the viability of cultured endothelial cells in the OGD assay compared to NIC, supporting a contention that cues adhering to ischemia inherent of RIC and BFRRE are important for the EV-induced protective effect. The protection of post-RIC EVs was further confirmed by measuring cell death after OGD. Interestingly, in this assay, pre-RIC EVs also exhibited protection of endothelial cells from OGD. This finding is in line with what has previously been shown in myocytes subjected to OGD, where naïve plasma EVs from rats and humans protected the cells through heat-shock protein 70 (Hsp70) [53]. With the cell-death assay, it was confirmed that a protective effect was still present when using pooled EVs from different volunteers, within the same group and time point. The rationale for pooling the EVs was first that the in vivo experiments required higher amounts of EVs per animal than per cell culture experiment. Second, testing the EVs from every single subject would increase inter-mice variability and would substantially increase the number of animals needed. As we observe a significant protective effect using pooled EVs in the OGD assay, we justify the use of pooled EVs for the in vivo experiment.

Despite the protection of endothelial cells by post-RIC EVs in the in vitro data, we did not observe a superior preservation of BBB integrity through the measurement of IgG extravasation at 4 h after reperfusion initiation in the tMCAO animals treated with post-RIC EVs. This may be because breakdown of the BBB in the early phase is mainly caused by the stress fiber formation and redistribution of junction proteins in the endothelial cells, partially because of MMP-9-mediated degradation of junction proteins [54], while loss of endothelial cells might not happen until later.

Protecting the microvasculature and improving angiogenesis after stroke have previously been suggested as being among the hallmarks of conditioning interventions. In a rat chronic cerebral hypoperfusion model, RIC was found to increase the number of vessels and angiogenesis in the hippocampus [55]. BFRRE was also reported to upregulate pro-angiogenic mRNAs such as vascular endothelial growth factor and inducible nitric oxide synthase [56,57]. In the current study, conditioned EVs isolated immediately after intervention did not significantly increase or decrease tube formation. We observed that tube-like structures would elongate and grow for the first 6 h after seeding the cells, and the tubes would either disintegrate or be maintained afterward. The tube structures that were maintained had fewer, but larger tubes at 24 h after seeding compared to the 6 h time point (Appendix A). Interestingly, for endothelial cells treated with post-BFRRE EVs, we observed a tendency toward continuing tube elongation even after the 6 h time point, pointing toward increased tube formation and stabilization at 24 h.

To further investigate if the protective effects occur in vivo, we employed a transient ischemic stroke mouse model. Interestingly, we observed a pronounced localization of post-RIC EVs in the ischemic hemisphere compared to the healthy hemisphere in tMCAO animals, indicating that RIC-conditioned EVs were homing to areas affected by ischemia-reperfusion. This corresponds with earlier reports in which EVs derived from neural stem cells and mesenchymal stem cells accumulated in the ischemic area in the brain [58,59]. Here, the accumulation of the post-RIC EVs in the ischemic area could be related to the inflammatory responses in the ischemic area. As suggested by Matsumoto et al., LPS can facilitate the brain uptake of EVs [60]. This is interesting as our in vitro data support that EVs can be taken up by endothelial cells and protect the cells against ischemia-reperfusion damage. EVs have been reported to impact physiological and pathological processes, including inhibition of immune response [61], inhibition of tumor growth [62], and stimulation of cell differentiation [63]. The surface protein components and lipid composition of EVs have been found to play a role in the targeting and signaling of EVs [64,65,66]. In cancer, it has been seen that the composition of integrins on tumor-derived EVs could decide the targeting organ of the EVs and initiate pre-metastasis at the target site [65]. In addition, fusion of specific ligands/peptides into EV membrane can achieve efficient EV targeting, such as fusion of neuron-specific RVG peptide leading to EVs targeting the brain [67], while fusion of ischemic myocardium-targeting peptide makes EVs target ischemic-injured cardiomyocytes [68]. In our study, the different behaviors of conditioned and non-conditioned EVs suggest that conditioning changes surface proteins on the EVs to facilitate targeting hypoxic areas. Future in vivo co-localization studies should investigate EV uptake by specific cells in the neurovascular unit, e.g., astrocytes, pericytes, and neurons, as well as activated microglia and infiltrating macrophages. What is noteworthy is that the leakage of BBB in the ischemic area or reduced clearance of post-RIC EVs from the brain might also contribute to the observed accumulation.

The size of the ischemic core increased substantially in the last 30 min of recording in the control group based on CBF changes from LSCI, but tended to remain unchanged in the EV-treated groups. This is consistent with the trend toward reduced infarct size following EV administration that was observed 4 h after reperfusion initiation as measured by Map-2 staining. Besides, with the longer investigation window of the 7-day reperfusion study, we observed no improvement in the neurological function or decrease in the infarct size 7 days after tMCAO. This is in contrast with the study from Li et al., where they similarly tested the effects of RIC-induced EVs in protecting the brain against cerebral ischemia [69]. The authors found that RIC EVs significantly reduced the infarct size in mice 24 h after a permanent MCAO induced by electrocoagulation. Our current results point to an earlier protection from post-RIC EVs in slowing down the development of the infarcted area. The non-significant reduction in infarct size observed in the present study in our transient MCAO model could be due to a combination of a rather short time window of investigation, the use of the transient MCAO stroke model, and/or type of anesthesia. Although these trends did not quite reach statistical significance, the mean values combined with our in vitro results and results on EV homing advocate that transient ischemic conditioning practiced as RIC can mediate acute tissue protective effects. Future experiments should seek to improve sample size on assessment of infarct size to provide a more robust conclusion. However, given the amounts of pooled blood required, we were not able to do this based on the blood material collected from the current underlying experiments. The differences induced by RIC EVs in the acute phase might have been more pronounced if the time course of measurements had been longer, which should be investigated in future studies. What is noteworthy is that compared to the study of Li et al., the reperfusion stroke model used in the current study was intended to mimic a reperfusion treatment in the clinic. A paradoxical side effect of reperfusion, known as reperfusion injury, causes addition damage after reestablishing perfusion in the penumbra [70,71]. Although the protective effect of RIC treatment has been proven in animal stroke models with reperfusion [72,73,74], the effective clinical trials were mostly done in patients who did not receive reperfusion treatment with either thrombolysis or thrombectomy [75,76]. Furthermore, a new multicenter clinical trial did not find decreased infarction by inducing RIC in patients receiving reperfusion treatment [77]. This points toward that reperfusion could attenuate the effect of RIC, especially in humans. The hemoglobin analysis from the MSRI showed that post-RIC EVs induced a higher oxyhemoglobin level in the ischemic core and penumbra area after reperfusion, which could suggest that more oxygen is available, thereby allowing for better tissue oxygenation in these areas compared to the control group. RIC has previously been found to improve oxygen supply in stroke mice as measured by a hypoxy probe after 30 min reperfusion from 60 min MCAO [78]. If the increased oxyhemoglobin at the early phase after reperfusion in the post-RIC EVs-treated group corresponds to better tissue oxygenation, EVs could be the mediators of the early beneficial effects of RIC. In addition to increased tissue oxygenation, the indication of acute tissue protective effects by post-RIC EVs could also be related with decreased oxidative stress by reducing the production of mitochondrial reactive oxygen species [79] or upregulating the Nrf2-ARE pathway, therefore increasing the expression of antioxidant proteins [14]. Besides, a reduced inflammatory response by suppressing NFκB activation and TNFα, decreased apoptosis by activation of STAT3 could also contribute to the effects of RIC [17]. Future studies could investigate the potential targets of conditioned EVs.

Also relevant, although EVs are usually thought of as non-immunogenic [80,81], transspecies transfer of EVs could exert some influence on the outcome of our experiments. However, the obvious advantage of using human plasma EVs for the current study is to standardize stimulatory protocols in ways that are not immediately possible in animal models. More specifically, we wanted to mimic ischemia reperfusion of RIC during loaded voluntary work (i.e., resembling high-pressure ischemia reperfusion of high-low and low-load resistance exercise) conducted to volitional failure. No current mouse models are able to offer involuntary resistance exercise to exhaustion. Moreover, unlike mouse models, human models easily allow for collection of serial blood samples of sufficient volume to allow analysis of biological effects (e.g., the MCAO model). Furthermore, we aimed to reduce inter-mice variabilities by using pooled EVs from heterogenic humans, which might introduce noise and cover the potential intervention effect in the in vivo experiments.

The present study is the first to compare the effects of EVs from differentiated ischemic +/− exercise conditioning interventions. Based on these comprehensive human experimental settings, which also included an individual non-intervention control group to account for potential confounding factors, both RIC and BFRRE EVs were shown to protect endothelial cells from OGD. Moreover, RIC stimulated in vivo homing of EVs to mice brain areas affected by ischemia via tMCAO and exhibited a tendency to slow down the development of infarction in stroke mice early after reperfusion, which could be related to increased oxygenation in the ischemic area. Reducing infarct growth in the early phase could expand the therapeutic window for reperfusion therapy in stroke patients. Currently, a large multicenter study is being conducted in stroke patients treated in a prehospital setting with RIC prior to reperfusion therapy [82]. Identifying the underlying mechanisms and potential benefits of RIC, BFRRE, and HLRE is pivotal because it may allow identification of protective molecules accessible for direct administration in the future. Additionally, targeted treatment with EVs delivering drugs directly to brain areas suffering from ischemia may facilitate high local concentrations of drugs in the affected area [83].

## 4. Material and Methods

### 4.1. Study Design

This study was conducted according to the standards of the Declaration of Helsinki after approval by the Ethical Committee for Region Midtjylland (ref. no. 1-10-72-218-16, 9 November 2016) and registered in the database clinicaltrials.gov (NCT03380663). Written informed consent was obtained from all participants. Forty-six participants were randomly assigned 1:1:1 to RIC, BFRRE, or HLRE (*n* = 12 in each group), or a non-intervention control group (NIC) (*n* = 10) in blocks of four in an open-label protocol (Figure 1). The total study period comprised 9 weeks, including a week with pre-assessment of basal capacity, a week with the single-bout session followed by recovery before a 6-week intervention period, and finally a week with post-training basal assessment. The standardization of visit procedures has previously been described [22,23], as well as the exclusion criteria. Briefly, the exclusion criteria were (1) resistance training within 6 months prior to inclusion, (2) participation in moderate/high intensity exercise training (other than resistance training) more than 1 h/week 6 months prior to inclusion, (3) smoking, and (4) use of prescription medication or intake of dietary supplements potentially affecting muscle metabolism and growth. Moreover, volunteers were instructed to maintain their habitual level of physical activity during the intervention period and to refrain from strenuous physical activity and alcohol for 3 days prior to all tests, including the single-bout session. The volunteers were asked to fast overnight prior to all test days.

### 4.2. Intervention Paradigms during Single-Treatment and Training

HLRE, BFRRE, and NIC were carried out as previously described [22,23]. In brief, HLRE consisted of four sets of 12 knee-extensions at 70% repetition maximum (RM) with 3 min inter-set recovery. BFRRE consisted of four sets of knee-extensions at 30% RM with 30 s inter-set recovery, with each set performed until volitional fatigue with occlusion provided by a 14 cm pneumatic cuff inflated to 50% of arterial occlusion pressure (AOP) as previously described [84]. RIC was performed using the autoRIC device (CellAegis Devices, Mississauga, ON, Canada) placed on one upper arm of the participant. The autoRIC automatically inflates and deflates a cuff, producing four cycles of 5 min temporary ischemia (200 mmHg) followed by 5 min of reperfusion (deflated cuff). During the 6-week intervention period, RIC, HLRE, and BFRRE were conducted three times per week, with the workload adjusted every other week to match improvement progression.

The NIC group underwent similar procedures, except the stimulation to control for effects of repeated blood sampling, nutrition control, and diurnal rhythm [41].

### 4.3. Blood Sampling

Blood samples were collected during the single-bout session before and after intervention. Blood sampling at the 30 min time point was based on previously unpublished pilot time course experiments, indicating that EVs were prone to increase and/or infer cell culture biological effects at that time point. While the 30 min time point provided a measure of the acute response to single treatment intervention, the 6-week time point (i.e., upon finalization of the prolonged repeated treatment/training protocol), allowed for a potential readout of chronic changes in the basal level of selected outcomes. An 18GA Venflon (BD Vialon, Becton Dickinson, Franklin Lakes, NJ, USA) was inserted into the cubital vein in the arm 30 min before collection of the first blood sample. Blood samples were collected in BD Vacutainer K2 EDTA tubes (Becton Dickinson) (see Figure 1 for the blood sampling time schedule). To obtain cell- and platelet-free plasma, the blood samples were centrifuged for 15 min at 1500× *g* at 20 °C in a swing bucket centrifuge (no brake). The plasma supernatant was transferred without disturbing the buffy coat to a new tube and centrifuged as before. Finally, the supernatant was cooled on ice and spun at 16,000× *g* for 10 min at 4 °C to remove platelets and cell debris, and then stored at −80 °C. Hemolysis was estimated by measuring the level of free hemoglobin at an absorbance of 414 nm. To avoid contamination of the EV preparations by lysed cells, plasma samples were discarded if A_414_ exceeded 0.2 [85]. From each of the four groups, six volunteers exhibiting low levels of hemolysis at all time points were included in the current study (anthropometrics, Appendix A).

### 4.4. EV Isolation and Characterization

EVs were isolated from 10 mL non-hemolyzed human plasma using size exclusion chromatography (SEC) using qEV10 columns (Izon Bioscience, Lyon, France) as described previously [34]. Only minor changes to the original protocol were made; dPBS was used to elute EVs for the in vitro assays. For each sample, fractions containing EVs without protein contamination were pooled and concentrated by ultrafiltration (Amicon Ultra-15, 10 kDa, Merck Millipore, Darmstadt, Germany) to a final volume of 2 mL. The EV size and concentration were determined by tunable resistive pulse sensing (TRPS) using a qNano Gold (Izon Bioscience), as previously described (Lassen et al., 2021), and by nanoparticle tracking analysis (NTA) by a Nanosight NS300 (Malvern Pananalytical, Malvern, UK). The samples (diluted 1:1000) were analyzed in five repeated measures (405 nm, camera level 14, syringe pump speed 20, threshold 5) and processed using Nanosight NTA software v. 3.0 (Malvern Pananalytical). The EVs were further visualized by transmission electron microscopy as previously described [34] and the presence of the canonical EV marker flotillin-1 was verified by Western blotting, as previously described [32].

### 4.5. Assessment of EV Phenotype by EV Array

The protein microarrays were produced in multitarget plates (MTP, Microfluor 2, 96 wells, polystyrene, Thermo Fisher Scientific, Waltham, MA, USA) and the printing was performed using a sciFLEXARRAYER S12 microarray printer installed with a Piezo Dispense Capillary (PDC) size 60 with coating type 3 (Scienion AG, Berlin, Germany). The printing procedure was performed under strict humidity and temperature control.

As positive controls, 30, 20, 10, 5, 3, 2, and 1 µg/mL of biotinylated goat anti-mouse IgG (Novus Biologicals, Littleton, CO, USA) were printed and PBS with 5% glycerol was used as negative control. After the print procedure, the MTPs were left to dry at room temperature (RT) overnight prior to further analysis. The 44 selected anti-human antibodies used for capturing are listed in Appendix A. All antibodies were diluted in PBS with 5% glycerol and printed in triplicate at 200 µg/mL. The EV Array was prepared as described [86] with modifications.

In short, the MTP was initially blocked (50 mM ethanolamine, 100 mM Tris, 0.1% SDS, pH 9.0) using a hand-held spray gun in a closed box for gentle application of the buffer. After 1 h of incubation, the EV array analysis was initiated by washing the MTPs in wash buffer (0.2% Tween20^®^ in PBS) using a HydroFlex™ microplate washer (Tecan Trading AG, Männedorf, CH). Then, a 50 µL sample and 50 µL wash buffer were added to each well and incubated for 2 h RT on an orbital shaker (450 rpm) followed by overnight incubation at 4 °C. After a wash procedure, each well of the MTPs was incubated with a 100 µL detection antibody cocktail (biotinylated anti-human-CD9, -CD63, and -CD81 (Ancell, Stillwater, MN, USA)) diluted 1:1500 in wash buffer for 2 h RT with shaking. Following a wash, 100 µL streptavidin-Cy3 ((Life Technologies, Waltham, MA, USA) diluted 1:3000 in wash buffer) was added to each well and incubated for 30 min RT on the shaker. The analysis was finalized by washing with wash buffer and subsequently with MilliQ water.

The MTPs were dried and scanned using a sciREADER FL2 microarray scanner (Scienion AG, Berlin, Germany), at 535 nm and an exposure time at 500 ms.

For differential expression analysis, the intensity values were log2 transformed and used as input for the R package LIMMA [87]. As continuous samples from the same subject at different time points were present, individual differences were blocked out using the duplicate correlation function. A surface marker was considered differentially expressed if the *p*-value was below 0.05.

### 4.6. EV Normalization Strategy and Pooling

For the in vitro assays, EVs isolated from each individual subject was used. We adjusted the EV concentration within each subject, so that the EV concentration used was adjusted to the blood collection time point with the lowest EV concentration within each subject. In this way, individual differences were respected and the same number of EVs was used for all timepoints within each subject (Appendix A).

For the in vivo and flow cytometry experiments, the EVs isolated from the individual volunteers, within the same intervention group, were pooled to provide the required number of EVs needed for in vivo injections in multiple animals or estimating cell death using flow cytometry. The EVs were pooled in the following way: for the in vitro flow cytometry assay, separate pools of pre-EVs and 5 min post-EVs were prepared from volunteers undergoing RIC. For the in vivo studies, all 5 min and 30 min post-EVs from volunteers undergoing RIC were pooled. Pre-EVs from the same volunteers were also pooled.

### 4.7. Cell Culture and Handling

Primary human brain microvascular endothelial cells (HBMECs) from ScienCell™ Research laboratories (Carlsbad, CA, USA, Catalog #1000) were used in passages 3–6 in all assays. Cells were maintained in complete Endothelial Cell Medium (ECM, ScienCell™ Research Laboratories) in a CO_2_ incubator at 37 °C, 5% CO_2_, and 95% humidity. Before seeding, all flasks and plates were coated in 2 μg/cm^2^ fibronectin bovine plasma (Sigma-Aldrich, St. Louis, MO, USA) and in Mg^2+^- and Ca^2+^-free dPBS. For testing, HBMECs were grown in T75 flasks, detached using Accutase (Sigma-Aldrich), and resuspended in 1 mL glucose-depleted DMEM (Glc(-) DMEM) (Lonza, Basel, Switzerland) before being seeded in the appropriate test plates.

### 4.8. EV Labeling and Uptake

For EV uptake by HBMECs, SEC-purified EVs were labelled using the PKH67 Green Fluorescent Cell Linker Kit (Sigma-Aldrich), as previously described [32]. After SEC separation, the sample was mixed 1:1 with ECM (ScienCell™ Research Laboratories). The no-EV control underwent the same procedure of labeling and SEC separation. HBMECs were seeded at 10,000 cells/well in µ-Slide VI 0.4 slides (ibidi, Gräfelfing, Germany) and incubated for 24 h before adding labelled EVs (10^9 EVs/mL), followed by an additional 24 h incubation. The cells were washed before fixation in 4% paraformaldehyde (PFA) and DAPI was used to stain nuclei. Images were obtained on a Leica fluorescent microscope (DM6000 B, Leica Microsystems, Wetzlar, Germany).

### 4.9. Measuring Cell Viability and Cell Death after Oxygen and Glucose Deprivation (OGD)

To mimic occlusion and reperfusion in vitro, HBMECs were subjected to an oxygen and glucose deprivation assay consisting of 4 h of oxygen deprivation followed by 4 h of normoxia to mimic reperfusion. For assaying cell viability, 10,000 cells/well passage 3 HBMECs were seeded in 96-well plates (white walled, clear bottom, Thermo Fisher Scientific). The HBMECs were maintained for 2 days until 90% confluency and washed in dPBS before adding purified EVs in dPBS mixed 1:1 with Glc(-) DMEM. As a positive control, 10 mM glucose was added in the oxygen deprivation (OD) control wells. After incubation for 45 min at 37 °C, 5% CO_2_, and 90% humidity (normoxic conditions), the plates without lids were moved to a hypoxia chamber (STEMCELL Technologies, Grenoble, France). The chamber was flushed for 45 min with anoxic gas (5% CO_2_, 95% N_2_) and HBMECs were incubated at anoxic conditions for 4 h in total. Then, the cells were returned to normoxic conditions and incubated for 4 h mimicking the reperfusion stage. Cell viability was determined using the ATP assay (CellTiter-Glo^®^ 2.0 Assay, Promega, Madison, WI, USA) according to the manufacturer’s protocol. The resulting luminescence was recorded by a Synergy HTX multi-mode reader (BioTek, Winooski, VT, USA). To directly measure cell death, HBMECs were seeded in fibronectin-coated six-well plates (100,000 cells/well) 2 days before performing the in vitro OGD assay described above. The cells were then harvested, washed to obtain a single-cell suspension, and kept on ice. Then, propidium iodide (10 μg/mL) was added just prior to flow cytometry analysis (Appendix A) on a Cell Sorter SH800 (Sony, Tokyo, Japan). The data was analyzed using FlowJo™ software (Becton Dickinson, Franklin Lakes, NJ, USA).

### 4.10. In Vitro Angiogenesis Assay

The tube formation assay was carried out according to the standard ibidi μ-slide Angiogenesis protocol (ibidi) using a Matrigel Growth Factor Reduced Basement Membrane Matrix (Matrigel, Corning, NY, USA). For each well, 6000 HBMECs were resuspended in 25 μL DMEM and mixed with 25 μL dPBS with diluted EVs (dilution described above (Appendix A)) before seeding in the well. For negative and positive controls, the HBMECs were resuspended in 50 μL dPBS and ECM, respectively. The slides were incubated at 37 °C, 5% CO_2_, and 95% humidity. Each biological sample was assayed in triplicates, and images of emerging tubes were captured at 2, 4, 6, and 24 h post seeding by phase contrast microscopy using a 4 × objective Zeiss Axiocam 105 color (Carl Zeiss Microscopy, Oberkochen, Germany). The images were processed by Fiji ImageJ (version 2.3.0), and total tube length was quantified by the Angiogenesis Analyzer plugin [88]. Pictures wrongly analyzed because of low contrast, dirt, or large over/underestimation of tubes were discarded.

### 4.11. Experimental Stroke Mouse Model

All experiments were carried out in accordance with the regulations of the Danish Ministry of Justice and Animal Protection Committees. The study was approved by the Danish Animal Experiments Inspectorate with permit number 2017-15-0201-01355 (15 January 2018) and complies with the ARRIVE guidelines 2009 (Animal Research: Reporting In Vivo Experiments). Seventy-five 8-week-old male C57BL/6NTac mice (Taconic, Denmark) were group housed 3–5 per cage in IVC cages with controlled temperature at 21 ± 1 °C and humidity at 50%. Animals were acclimated at least 2 weeks and handled before all experiments. Focal ischemic stroke was induced in mice by transient middle cerebral artery occlusion (tMCAO) [89]. Briefly, the mouse was anesthetized with ketamine (60 mg/kg) and xylazine (10 mg/kg) and endotracheal intubated to provide mechanical ventilation. The body temperature was kept at 37 ± 0.5 °C with a rectal probe and heating pad. A tail vein catheter was inserted for later EV injection. For the non-survival in vivo imaging experiments, the left femoral artery was cannulated for blood pressure monitoring and blood sampling for blood gas measurements (Appendix A). The left common carotid artery (CCA) and external carotid artery (ECA) were carefully dissected and ligated without affecting the vagus nerve. A microvascular clip was used to close the left internal carotid artery (ICA) temporarily. Then a small incision was made in the CCA, through which a silicone-coated monofilament (602256PK5Re, Doccol, Sharon, MA, USA) was inserted to block the middle cerebral artery (MCA) in the circle of Willis temporally. After 45 min occlusion, the filament was withdrawn for reperfusion. For the 7-day reperfusion study, mice were sutured after reperfusion initiation and transferred to a recovery chamber at 32 °C. Fluid supply was given by subcutaneous injection of 1 mL 0.9% saline. After 1 h in the recovery chamber, mice were returned to their home cage and supplied with soaked fodder and water. Analgesic was given by subcutaneous injection of buprenorphine (Temgesic, 0.1 mg/kg/8h) before surgery and three times during the first 24 h after surgery.

### 4.12. In Vivo EV Tracking

To visualize EV distribution in the brain after a system injection, RIC EVs were labeled with the ExoGlow^TM^-Vivo EV Near IR labeling kit (SBI, Palo Alto, CA, USA) according to the supplier’s protocol. TMCAO mice were injected with either labeled EVs (pre-RIC EVs, *n* = 3; post-RIC EVs, *n* = 4) or negative control (Krebs–Henseleit (KH) buffer, *n* = 4), which underwent the same labeling process as EVs, via the tail vein catheter at 20 min after occlusion onset. After reperfusion initiation, the wound was sutured and animals were transferred to a recovery chamber with controlled temperature (32 °C), supplied with water and soaked fodder. Four hours after reperfusion, mice were euthanized with a pentobarbital sodium overdose (Exagon vet., Richter Pharma, Wels, Austria). Brains were taken out and scanned using an Odyssey Sa Infrared Imaging System (LI-COR, Lincoln, NE, USA) with a focus of 37.5 mm, resolution of 20 μm, and excitation laser of 785 nm. For the analysis of EV accumulation in the ischemic brains, regions of interest (ROIs) with the same size (1 mm^2^) were placed in the area supplied by the MCA in the ischemic hemisphere and contralateral heathy hemisphere. Mean fluorescent intensity in the ROIs were measured by Fiji ImageJ (version 2.3.0). To exclude the possibility of different labeling efficiency between pre- and post-RIC EVs, the fluorescent intensity in the ROIs of the ischemic hemisphere were normalized to the corresponding ROIs in the healthy hemisphere before between-group comparisons.

### 4.13. Cerebral Blood Flow (CBF) and Hemoglobin Measurement

To record the acute overall CBF and hemoglobin oxygenation changes over time during and following the occlusion, mice were subjected to both laser speckle contrast imaging (LSCI) [90] and multispectral reflectance imaging (MSRI) [91]. After filament insertion during the tMCAO procedure, the animal was turned over and head fixed in a stereotaxic frame under the camera of the moorO_2_Flo (Moor Instrument, Axminster, UK), which combines LSCI with MSRI. Eye gel (Viscotears, Bausch & Lomb Nordic AB, Stockholm, Sweden) was added to the exposed skull to keep it moisturized for improving translucency during the imaging process. Imaging was started as soon as possible after occlusion and kept continuously recording until 4 h after reperfusion initiation. The experimenter was blinded to the treatments and the mice were randomly given either pre-RIC EVs (*n* = 8), post-RIC EVs (*n* = 9), or KH-buffer as the non-treated control (*n* = 7). Among them, two mice were excluded from the post-RIC EVs group because of hemorrhage during surgery and disrupted imaging, respectively. Blood gas measurements (ABL90 FLEX blood gas analyzer, Radiometer, Copenhagen, Denmark) were performed on arterial blood samples taken from the femoral artery catheter at three time points, before tMCAO, 2 h, and 4 h after reperfusion initiation to monitor animal health during the procedure (Appendix A). Fluid supply was provided by subcutaneous injection of 1 mL saline at the beginning of surgery and in the middle of imaging. After taking the final blood sample, mice were euthanized with pentobarbital. Brains were taken out and snap frozen at −50 °C in isopentane cooled by dry ice.

### 4.14. In Vivo Image Analysis

The imaging data from moorO_2_Flo was recorded with a sampling rate of one frame per second. For data analysis, the sampling rate was reduced to one frame per minute and analyzed with in-house build Matlab scripts. Frames that were marked with EV injection and reperfusion during recording were noted for each animal to align the respective time points in the results. Before CBF analysis, the big vessels were masked out by thresholding and the mean flow intensity in the healthy hemisphere before reperfusion were used as baseline. To track the temporal development of the ischemic core, pixels in the ischemic hemisphere with flow intensities less than 33% of baseline were defined as the area of the ischemic core [92]. In this way, the number of pixels in the ischemic core over time was calculated. To estimate the perfusion and hemoglobin changes in three areas over time, area masks for the ischemic core (<33%), penumbra (33–70%), and normal perfused area (>70%) were defined based on the mean flow in the ischemic hemisphere before reperfusion. The relative flow, oxyhemoglobin (oxyHb) and deoxyhemoglobin (deoxyHb) changes in these areas were analyzed during the 4 h reperfusion period. All the relative changes were scaled to start at 1 from the beginning of the data acquisition.

### 4.15. Measuring the Effects of EVs on Neurological Function in the tMCAO Mice

To evaluate whether RIC EVs can provide prolonged neuroprotection after prolonged reperfusion (7 days after stroke), three doses of pre-RIC EVs (*n* = 14), post-RIC EVs (*n* = 13) (about 1 × 10^10^ EVs in 250 µL KH-buffer for each injection), or KH-buffer (*n* = 13) were injected intravenously during occlusion, at day 3 and 5 after tMCAO surgery. As two mice from each group died during recovery after tMCAO surgery, we ended up with *n* = 12 for pre-RIC EVs, *n* = 11 for post-RIC EVs, and *n* = 11 for vehicle control group. For every day of neurological function assessment, a five-point scale was used as described before [93,94]. No neurologic deficit gets a score of 0; failure to extend right forepaw fully gets a score of 1 (mild focal neurological deficit); circling to the left gets a score of 2 (moderate focal neurological deficit); falling to the left get a score of 3 (severe focal neurological deficit); and no spontaneous walk and decreased level of consciousness get a score of 4. A score between 1–4 was taken as a successful induction of tMCAO.

### 4.16. Behavioral Tests

The corner test measures the sensation of vibrissae and motor functions of mice using a home-made setup consisting of a large 10 mm thick plexiglass sheet that is folded, forming an angle of 30° with sides 30 × 20 cm [95,96]. After left-side tMCAO surgery, mice have deficits in the contralateral side vibrissae sensations and motor functions, which increase the frequency of turning to the right, thus giving a measure of the level of deficiency after tMCAO. A corner test was conducted before surgery and 3 days after tMCAO with 10 trials for each mouse to calculate the frequency of turning right. Hargreave’s test measures the sensation of thermal pain of the mouse [97]. The apparatus automatically records withdrawal latency of the hind paw from the heating source (Ugo Basile, Gemonio, Italy). If tMCAO surgery leads to a deficiency of thermal pain sensation in the contralateral hind paw, it will result in longer withdrawal latency [98,99]. Hargreave’s test was conducted before surgery and 7 days after tMCAO surgery. The mice were allowed to acclimatize for 15 min in the test apparatus before the measurement. Each hind paw was measured in five trials, with an interval of 20 min between each trial. The withdrawal latency was calculated as the mean value of the three trials after removing the highest value and the lowest value measured in the five trials [97]. After the Hargreave’s test on day 7, mice were sacrificed, and the brains were frozen as described above for cryosectioning and staining.

### 4.17. Brain Tissue Cryosectioning and Immunohistochemistry (IHC)

Frozen mouse brains were attached to the tissue holder with water to avoid cracking during sectioning with a cryostat. The temperature of the knife and the holder in the cryostat was set to −12 °C, while the cabinet was at −20 °C. Six series of 20 µm sections were collected every 600 µm and mounted on Superfrost Plus glass slides (Thermo Fisher Scientific) and stored at −80 °C.

For IHC staining, the frozen sections were thawed and air dried at RT for 15 min followed by post-fixation in 4% PFA for 15 min. Excess PFA was removed with a tris buffered saline (TBS) rinse. Microtubule-associated protein 2 (Map2), which is rapidly degraded in dendrites following ischemia, was used to assess infarct volumes [100], while the potential leakage of the BBB was measured by co-staining for CD31 (endothelial cell marker) and mouse immunoglobulin G (IgG). IgG is a serum protein that exists in the blood and extravasation of IgG outside the vessels in the brain is an indicator of BBB damage. For Anti-Map2 staining, a 22h antigen retrieval with citrate buffer at 60 °C was performed. Endogenous peroxidases were blocked with 3% H_2_O_2_. The sections were permeabilized with 0.3% Triton-X-100 in TBS for 10 min and blocked with 2% BSA for 1 h. The slides were incubated with primary antibody rabbit anti-Map2 (1:800, ab32454, Abcam, Cambridge, UK) overnight at 4 °C in a moisturized chamber. After three 15 min TBS rinses, the slides were incubated with diluted polyclonal secondary antibody goat anti-rabbit IgG/HRP (1:400; P0448, Agilent, Glostrup, Denmark) for 2 h. After visualization of bound antibody with DAB (3,3′-diaminobenzidine), slides were mounted with DPX mounting medium (Hounisen, Skanderborg, Denmark). Fluorescent staining was done for co-staining against CD31 (Rat anti-CD31, 1:50, Catalog No.550274, BD Biosciences, Herlev, Denmark) and mouse IgG (Biotinylated horse anti-mouse IgG, 1:400, Vectorlabs, Søborg, Denmark) using the same protocol without antigen retrieval and endogenous peroxidase blocking. Goat anti-rat IgG conjugated with Alexa Fluor 488 (1:400, A-11006, Thermo Fisher Scientific) and Alexa Fluor 546 streptavidin (1:400, S11225, Thermo Fisher Scientific) were used as secondary antibodies, respectively. Antifade mountant (Thermo Fisher Scientific, P10144) was used to mount the fluorescent slides with coverslip, which were sealed with nail polish and protected from light until imaging.

### 4.18. Infarct Volume Estimations

Infarct volumes were estimated based on anti-Map2-stained serial sections using the Cavalieri estimator [101] in the NewCAST software version 4.4.5.0 (Visiopharm A/S, Hørsholm, Denmark). Each section was visualized using a microscope (Olympus BX50, Ballerup, Denmark), and the infarct area was estimated using a 2D nucleator by a blinded researcher. Total infarct volumes were calculated using the formula: V = d·Σa_i_ (d = distance between sections, a_i_ = cross sectional area) [102]. Due to complications in sectioning and staining of the brains, the number of animals included in the infarct estimations for the acute study were pre-RIC EVs, *n* = 8, post-RIC EVs, *n* = 5, and controls, *n* = 6, and for the 7-day reperfusion study were pre-RIC EVs, *n* = 7, post-RIC Evs, *n* = 8, and controls, *n* = 10.

### 4.19. Imaging and Analysis of the IHC Sections

For the imaging of anti-CD31 and anti-IgG co-stained sections, three sections with the largest infarction areas were picked for fluorescent microscopy (Zeiss AXIO Imager M2, Zeiss, Germany) for each animal. For the analysis of IgG extravasation, three to four ROIs were picked in the infarct area, penumbral area (area adjacent to the infarction), and healthy area, respectively for each section. All ROIs were analyzed by Fiji ImageJ (version 2.3.0) with the same settings to get CD31 and IgG-stained areas. ROIs were averaged for each section to get the CD31 and IgG areas in different regions. IgG extravasation was calculated as the area of IgG divided by the area of CD31 for each section in different regions.

### 4.20. Statistics

EV concentration and size measurements from the NTA data were analyzed using mixed-effect model (REML) analysis with no assumption of sphericity and implementing Geisser-Greenhouse correction. To correct for multiple comparisons, Dunnett’s test was applied. The OGD and angiogenesis assay were analyzed using repeated measures with two-way ANOVA with no assumption of sphericity and implementing Geisser-Greenhouse correction. For the within-group comparisons, all post-intervention timepoints were compared to the pre-intervention timepoint. For the inter-group comparison, all post-intervention timepoints were normalized to the pre-intervention timepoint, within each subject, and then the post-intervention timepoint was compared to the NIC. To correct for multiple comparisons, Dunnett’s test was applied. The cell death assay measured by flow cytometry was analyzed using a one-way ANOVA. To correct for multiple comparisons, Sidak’s test was applied. For the in vivo data, a one-way ANOVA with Tukey’s multiple comparisons test or a two-way ANOVA with Sidak’s multiple comparisons test was performed. Mixed-model analysis was conducted for the imaging data. For all tests, a p-value below 0.05 was considered significant. The statistical analyses were performed in GraphPad Prism (Version 8.4.0, San Diego, CA, USA).

## Figures and Tables

**Figure 1 ijms-23-03334-f001:**
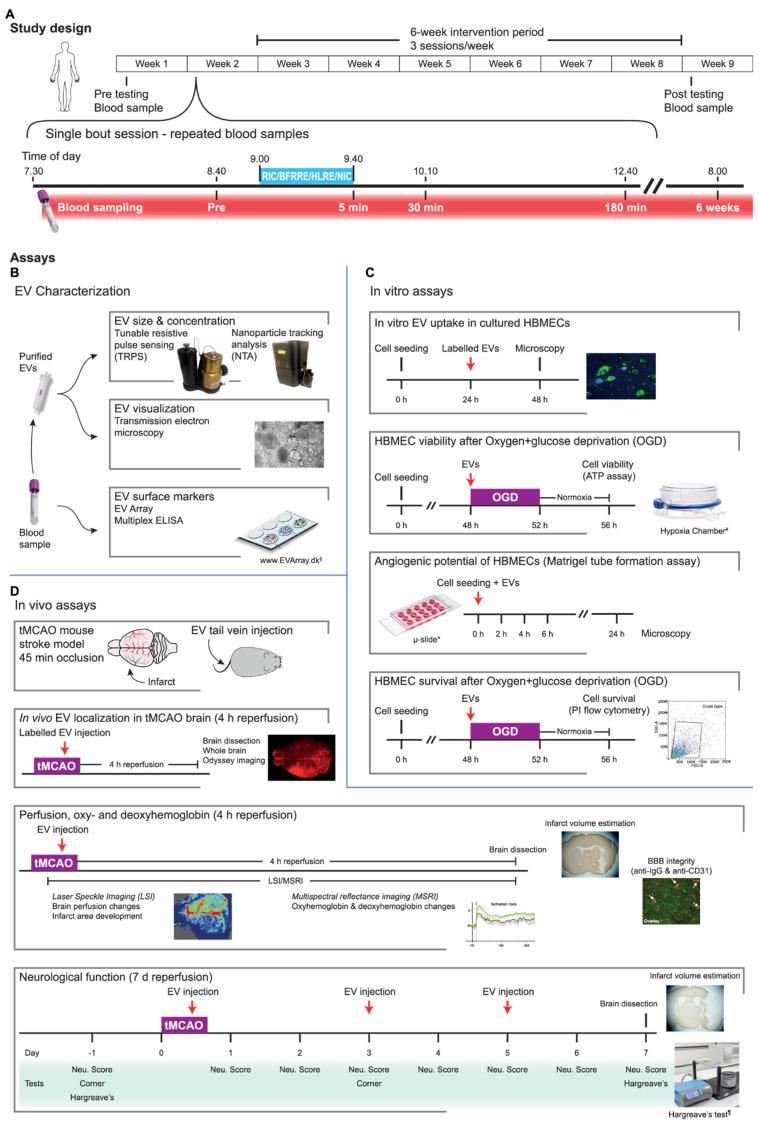
Schematic overview of the full study design including intervention period, characterization of extracellular vesicles (EVs), and in vitro and in vivo assays. The study design (**A**) depicts the intervention period including pre- and post-blood sampling timepoints. The human participants were randomized to either of the three conditioning groups (remote ischemic conditioning (RIC), blood-flow-restricted resistance exercise (BFRRE), or high-load resistance exercise (HLRE)), or a non-intervention group (NIC) that underwent the same blood-sampling regime. The timing of the repeated blood sampling is shown in the single-bout session. All participants started at the same time of day to minimize variations from diurnal rhythm. EV characterization (**B**): surface markers on plasma EVs were characterized by EV Array multiplex ELISA analysis. To obtain concentration and size distribution, the purified EVs were analyzed using tunable resistive pulse sensing (TRPS) and nanoparticle tracking analysis (NTA). EVs were visualized using transmission electron microscopy (TEM). In vitro assays (**C**): all in vitro assays were carried out in primary human brain microvascular endothelial cells (HBMECs). We tested EV uptake, viability, and cytotoxicity after oxygen-glucose deprivation (OGD), and angiogenic potential (tube formation assay). In vivo assays (**D**): for the in vivo studies, a transient middle cerebral artery occlusion (tMCAO) mouse model was used to assess EVs in a stroke model. Accumulation of EVs was assessed by whole-brain imaging of fluorescently labeled EVs. Brain perfusion was assessed with laser speckle imaging (LSI), while multispectral reflective imaging (MSRI) was performed to obtain changes in brain oxygenation. IgG extravasation was used to assess blood-brain barrier integrity. To assess stroke severity, behavior tests (corner and Hargreaves tests) and neurological scoring were performed. Reproduced with permission from Malene Møller Jørgensen, www.EVArray.dk, accessed on 1 March 2022. Courtesy of STEMCELL Technologies, Grenoble, France; *ibidi GmbH, Gräfelfing, Germany; and Ugo Basile SRL, Gemonio, Italy.

**Figure 2 ijms-23-03334-f002:**
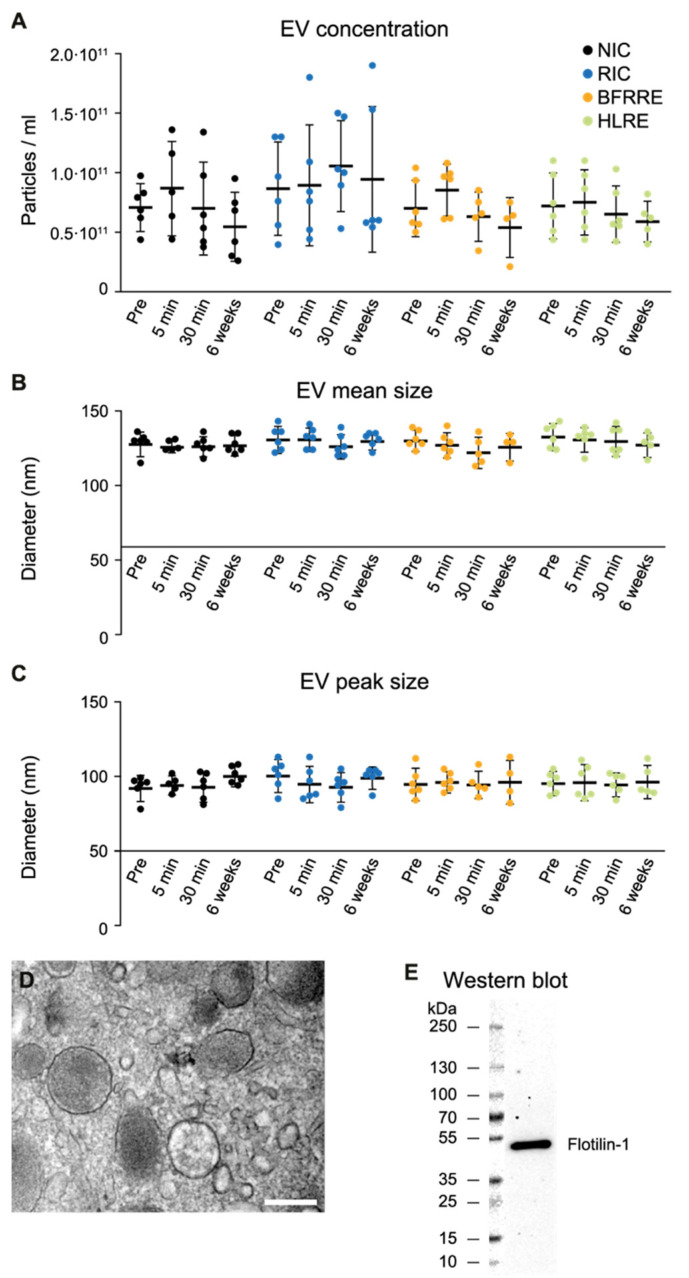
Characterization of extracellular vesicles (EVs) isolated from plasma before and at different timepoints after intervention. The average EV concentration and EV size in remote ischemic conditioning (RIC), blood-flow-restricted resistance exercise (BFRRE), high-load resistance exercise (HLRE), and non-intervention control (NIC) samples (*n* = 6 for each group, before intervention (pre) and after intervention (5 min, 30 min, and 6 weeks) (**A**–**C**). Neither the EV concentration, nor the EV size, changed in the RIC, BFRRE, HLRE, or NIC group. Epoxy-embedded isolated EVs were visualized by transmission electron microscopy (**D**). Scale bar equals 200 nm. Vesicles ranging in size from 50–200 nm were observed. The presence of the canonical EV marker Flotillin 1 on isolated EVs was validated using Western blotting (**E**). Data are presented as mean ± SD.

**Figure 3 ijms-23-03334-f003:**
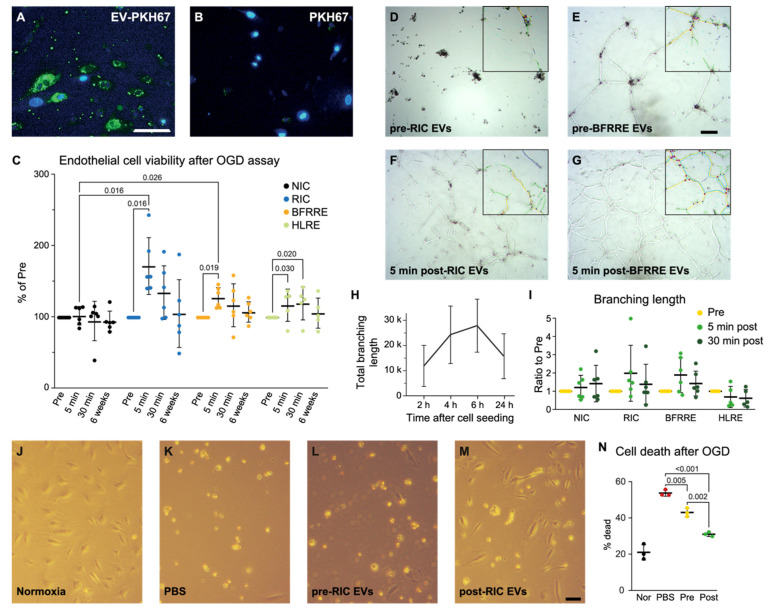
Conditioned extracellular vesicles (EVs) released after remote ischemic conditioning (RIC), blood-flow-restricted resistance exercise (BFRRE), and high-load resistance exercise (HLRE) improved viability of human brain microvascular endothelial cells (HBMECs) after oxygen and glucose deprivation (OGD) but did not promote tube formation of HBMECs. Conditioned EVs were added to HBMECs undergoing OGD. The EVs were internalized by HBMECs (**A**), while no internalization of dye alone was observed (PBS treated with dye and purified as the EVs (**B**)). Scale bar equals 50 µm. The viability of EV treated HBMECs, as measured by the level of ATP, was assessed after 4 h of OGD and 4 h of reperfusion (**C**). Post-RIC EVs (5 and 30 min), post-BFRRE EVs (5 min) and post-HLRE EVs (5 and 30 min) protected HBMECs from global OGD compared to pre-EVs. Furthermore, post-RIC EVs (5 min) and post-BFRRE (5 min) increased viability compared to post-EVs (5 min) in the NIC group. EVs from the remaining timepoints had no effect (**C**). Data are presented as mean ± SD (samples from six volunteers in each group). *p*-values are shown for significant comparisons. Representative phase contrast microscopy images showing tubes 24 h post seeding for HBMECs treated with pre-RIC EVs (**D**), post-RIC EVs (**F**), pre-BFRRE EVs (**E**), and post-BFRRE EVs (**G**). Box in the upper righthand corner indicates branch length analyzed by the angiogenesis analyzer plug-in for ImageJ. Scale bar equals 200 µm. HBMECs cultured at optimum conditions show a peak of branching length 6 h after seeding (**H**). None of the conditions showed a significant change in tube formation, in part because of high variance. Compared to pre-intervention EVs, HBMECs treated with post-RIC EVs (5 min) or BFRRE EVs (5 min) showed a non-significant higher branching length compared to pre samples after 24 h (**I**). Data are presented as mean ± SD (samples from six volunteers in each group). Morphology and cell survival of HBMECs treated with conditioned EVs and subjected to OGD (**J**–**M**). A marked difference in the morphology of the HBMECs was observed after incubation at normoxic conditions (**J**) compared to the OGD-treated HBMECs, PBS (control, (**K**)), pre-RIC EVs (**L**), or post-RIC EVs (5 min, (**M**)). Scale bar equals 100 µm. The percentage of propidium iodide-positive HBMECs, marking cell death, was reduced when treated with pre-RIC EVs and further reduced when treated with post-RIC EVs (**N**). Data are presented as mean ± SD. *p*-values are shown for significant comparisons.

**Figure 4 ijms-23-03334-f004:**
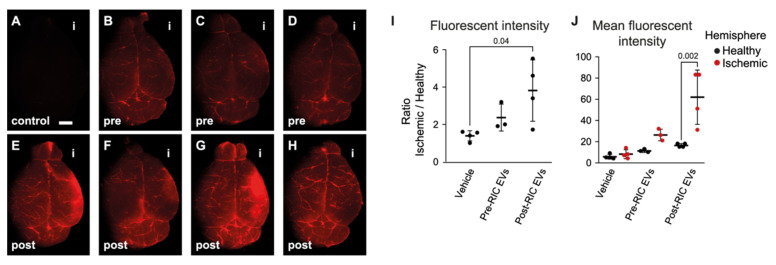
Accumulation of labelled extracellular vesicles (EVs) in murine brains following transient middle cerebral artery occlusion (tMCAO). Fluorescent brain images of single tMCAO mice injected with vehicle ((**A**), representative negative control labelling without EVs, *n* = 4), pre- remote ischemic conditioning (RIC) EV mice ((**B**–**D**), *n* = 3), and post-RIC EV mice ((**E**–**H**), *n* = 4), “i” indicating the side of ischemia. Scale bar, 2 mm. After normalization of the fluorescent intensities to the healthy hemisphere in each animal, a significant accumulation of EVs was seen in post-RIC EV-treated animals compared to the vehicle control (**I**). Post-RIC EVs accumulated in the ischemic hemisphere when comparing the mean fluorescent intensity in the region of interests (ROIs) from the ischemic hemisphere and the healthy hemisphere (**J**). Data are presented as mean ± SD. *p*-values are shown for significant comparisons.

**Figure 5 ijms-23-03334-f005:**
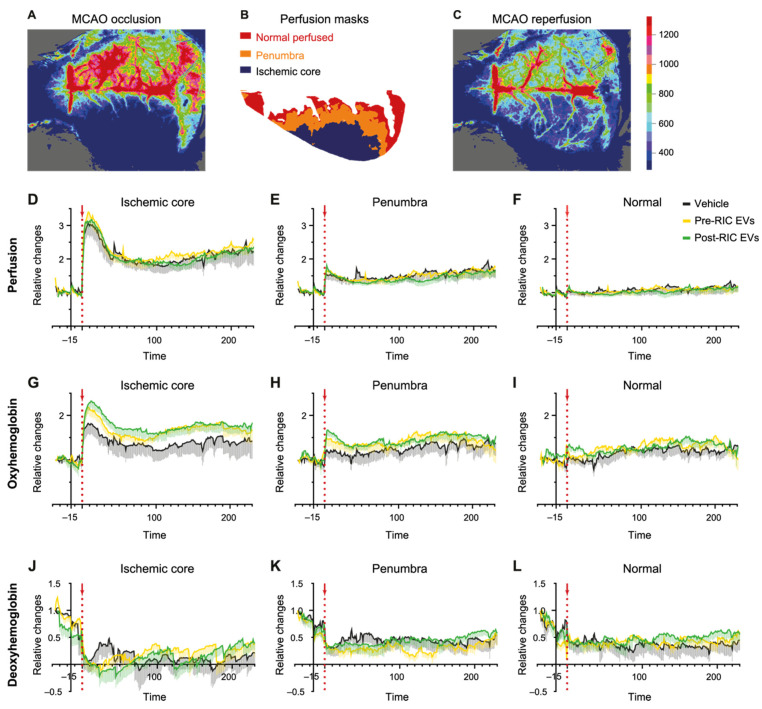
Laser speckle contrast imaging of perfusion, oxyhemoglobin, and deoxyhemoglobin changes in the mouse brain during stroke and reperfusion. Laser speckle images of the perfusion during occlusion (**A**) and in the reperfusion stage (**C**). Analysis masks (**B**) defining the ischemic core, penumbra, and normal perfused areas based on the level of cerebral blood flow reduction used for analyzing the changes in perfusion (**D**–**F**), oxyhemoglobin (**G**–**I**), and deoxyhemoglobin (**J**–**L**). Vehicle or pre- or post-remote ischemic conditioning (RIC) extracellular vesicles (EVs) were injected during occlusion 15 min before reperfusion (the red arrow marks the time of reperfusion, aligned at 0 min). Data are presented as mean ± SEM.

**Figure 6 ijms-23-03334-f006:**
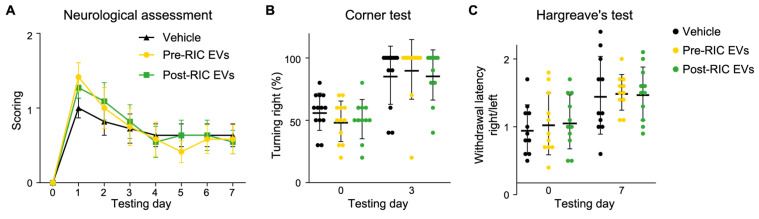
Neurological function measurements following repeated extracellular vesicle (EV) administration in transient middle cerebral artery occlusion (tMCAO) mice. Stroke mice received either pre-remote ischemic conditioning (RIC) EVs (*n* = 12), post-RIC EVs (*n* = 11), or vehicle (*n* = 11) during occlusion and at day 3 and 5. A score of their neurological function was given every day (**A**) showing slow recovery of the tMCAO mice after day 1. Corner test showed no turning preference before surgery while the mice primarily turned to the right as expected (**B**). As expected, the withdrawal latency after heat stimulation was increased in the right hind paw following tMCAO (**C**). The EV treatment did not show an effect in any of the measures. Data are presented as mean ± SD.

**Figure 7 ijms-23-03334-f007:**
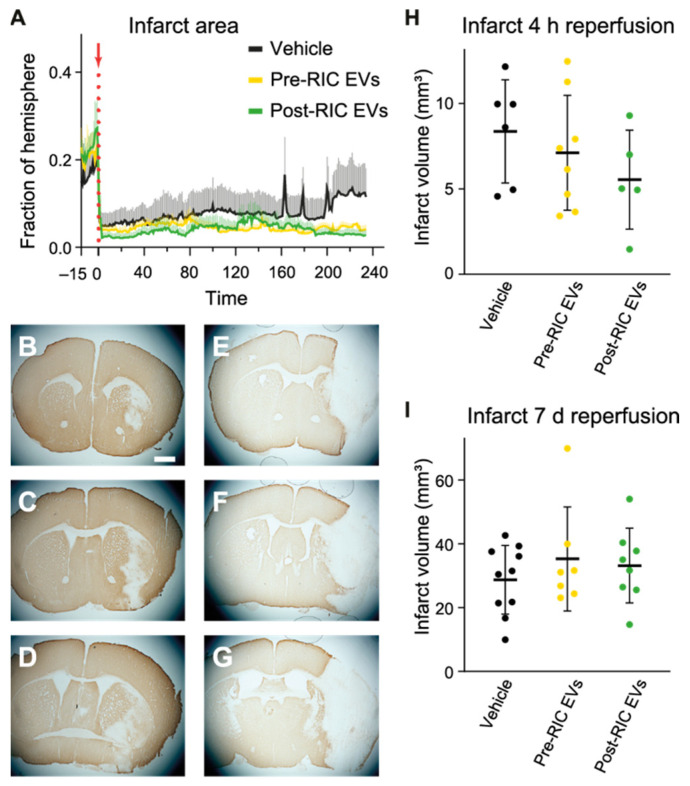
Ischemic core development during occlusion and reperfusion and final infarct volume in mice subjected to transient middle cerebral artery occlusion (tMCAO). Laser speckle contrast imaging of the ischemic core during the first 4 h of reperfusion ((**A**), number of pixels with flow intensity less than 33% of baseline cerebral blood flow, (the red arrow marks the time of reperfusion, aligned at 0 min)). Infarct volumes at termination were assessed based on anti-Map-2 staining showing early infarction in the striatum after 4 h reperfusion ((**B**–**D**), adjacent sections from a post-remote ischemic conditioning (RIC) EV treated animal), which develops into a large cortical infarct after 7-day reperfusion ((**E**–**G**), adjacent sections from a post-RIC EV treated animal). Scale bar, 1 mm. Mean and individual infarct volumes in tMCAO animals after 4 h reperfusion (**H**) and 7-day reperfusion (**I**). Data are presented as mean ± SD.

## Data Availability

Data will be made available upon request.

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
