# Peer review of "The Role of Plasma Extracellular Vesicles in Remote Ischemic Conditioning and Exercise-Induced Ischemic Tolerance"

_ijms, 2022, doi:10.3390/ijms23063334_

Round 1
Reviewer 1 Report
Gu et al suggested that circulating EVs induced by RIC and BFRRE can mediate protection, but the in vivo and translational effects of conditioned EVs require further experimental verification. This study was well designed, results were sound. However, there are still several issues that need to be addressed.
- Human blood was collected in this study, please provide the ethical approval information.
- Please provide scale bars of all photos.
- Please discuss the clinical implications of this study.
- Please discuss the limitations of this study.
Author Response
Reviewer 1 comments
Comments and Suggestions for Authors
Gu et al suggested that circulating EVs induced by RIC and BFRRE can mediate protection, but the in vivo and translational effects of conditioned EVs require further experimental verification. This study was well designed, results were sound. However, there are still several issues that need to be addressed.
1. Human blood was collected in this study, please provide the ethical approval information.
The ethical approval is included in the Method section (4.1 Study design, page 17, L643) as well as in the Institutional Review Board Statement (page 24, L1084-5)
2. Please provide scale bars of all photos.
Scalebars have been added in the photos of figure 4 (page 9) and figure 7 (page 7)
In general, in series of panels with the same magnification, scalebars are only shown on one of the panels
3. Please discuss the clinical implications of this study.
The clinical implications of the study have been added in the end of the discussion (page 16, L626-32)
4. Please discuss the limitations of this study.
The limitations of the study are included in the Discussion (page 15, L573-8; page 16 L604-6+L614-6)
Reviewer 2 Report
See attached PDF for complete comments.

Author Response
Reviewer 2 comments
General Comments:
The authors are to be commended for their hard work in attempting to quantify the effect of extracellular vesicles on ischemia tolerance. This could lead to therapies in which synthesized EVs could be administered to treat strokes or failing hearts. While worthy of publication, this manuscript needs some revision to meet the IJMS standard. In general, the authors should address the following specific comments below. Generally, the language of the paper is sufficient.
While disappointing, the mostly negative results could be bolstered with molecular data (protein, qRT-PCR data, etc.) to give a snapshot of the microenvironment that could be compared to other or future studies. Such a paper would be a good contribution to untangling the mystery of EV utility in therapeutic development.
Specific Comments:
Introduction
- L42-44: This sentence is awkward because it does not explain the cause of ischemic stroke. Please rephrase to indicate the cause (vascular lesion, blood clot, etc.) and then list your effects as deleterious effects of the infarction.
The cause of an ischemic stroke has been expanded upon: “Ischemic stroke is caused by restriction of blood supply to an area causing a shortage of oxygen and nutrients, which is usually a consequence of blood clot formation in the artery or an embolus traveling from other parts of the body e.g., the heart. The occlusion of blood flow leads to the failure of energy-consuming ion pumps and intracellular accumulation of Na+ and Ca2+. Overloading of intracellular Na+ and Ca2+ causes mitochondrial dysfunction and can trigger apoptosis or necrosis. Mitochondrial dysfunction also leads to an overproduction of reactive oxygen species (ROS) that induces oxidative stress. In all, this results in immediate cell death in the ischemic core” (page 1, L42-55)
- L44-46: You must clearly define ischemia with a biochemically suitable phrase, such as (for example) “in the core area of ischemia, due to oxygen deprivation that results in increased reactive oxygen species, whereas…” Ischemia itself is a complex phenomenon and must be carefully defined.
The description of ischemia has been rephrased to make it more biochemically suitable. (page 1, L42-55)
- L54-77: This section must contain at least SOME mechanistic suggestions as to why RIC works so well. Gene names or pathways need to be here to educate the reader. PMID: 32121587 may be a good reference for this.
Thank you for the suggestion. We have added some mechanism background of RIC to better explain RIC: “The working mechanism underlying RIC has been studied by multiple preclinical studies from different aspects, including the decrease of oxidative stress by initiating the antioxidant Nrf2-ARE pathway, regulation of inflammatory responses by blocking NFκB, upregulation of autophagy through IL-6 dependent JAK-STAT pathway” (page 2, L68-72)
- L78-91: It would be better to use an authoritative definition of EVs to prevent confusion with microvescles and other size categories. Suggest https://doi.org/10.1093/cvr/cvx211 as a good reference.
Thank you for pointing this out and suggesting a good reference. We have updated the description based on the reference and following the guidelines of the International Society of Extracellular Vesicles and their publication (MISEV 2018): “EVs is the collective term for small vesicles (diameter of 30nm to 1000nm) that includes exosomes, microvesicles, ectosomes, apoptotic bodies etc. They are unable to replicate and are encased by a lipid bilayer membrane that makes them stable in the extracellular space and body fluids” (Page 2, L91-4).
- L96-97: Although the word “subject” is still widely used in research reports, the words “volunteer” or “participant” are preferred. “Subject” carries the connotation of “forced and unwilling.”
We acknowledge the connotation of the word “subjects” and have replaced it with “volunteers” throughout the manuscript
- Additionally, which cells or organelles secrete EVs and how is this process mediated? Without molecular mechanisms here, the reader is unable to judge why exercise alone is sufficient to increase EV concentration. Additionally, if EVs do not induce an immunogenic response, then this needs to be made clear in order to justify the model (human EVs in mice).
We have included a description of EV secreting cells in the Introduction (page 2, L97-100) as well as a discussion of cross species immunogenicity in the Discussion (page 16, L605-7)
Results
- Figure 1: Did any of the measured parameters reach significance? If so, this needs to be clearly indicated. Any judgement on the effectiveness of the interventions with regard to EV concentration cannot be done without such information. Also, it seems as if EV concentration drops below baseline 6 weeks later in NIC, BFRRE, and HLRE groups. Why is this? If these changes in concentration are not significant, then neither is the intervention on EV concentration. Such tight control needs to be placed in context….is the intervention supposed to generate extra EV or make the present ones more adept at preventing ischemic damage? This information needs to be put in the Discussion.
We thank the reviewer for this comment. We agree that the results on EV concentration and size, measured before and after interventions, should have been more clearly stated - and should have been more thoroughly discussed. None of the measured parameters, concentration nor size, reached significance. Based on previous studies from other groups (reviewed in discussion), we expected to observe an acute (5 min after intervention) increase in the concentration of circulating EVs after RIC, BFRRE and HLRE. On the other hand, it was expected that we did not observe an increase in EV concentration at 30 min and 6 weeks after intervention (based on similar studies conducted by our group and taking into account the fast turnover time of EVs in the circulation). We can only speculate on the discrepancy on why some studies show an increased number of EVs in the circulation after e.g., RIC or exercise and why other studies do not. This could be a technical issue related to the use of size exclusion chromatography vs other EV isolation methods such as ultracentrifugation or precipitation, in that SEC depends on when the EVs elute from the column and what fractions are collected. However, we can conclude that using these intervention forms (RIC, BFFRE, HLRE) and using SEC to purify EVs, then we do not observe an increase in circulating EVs immediately after intervention. Thus, we believe, as the reviewer mentions, and as our in vitro results show, that the EVs do not prevent ischemic damage because of increased numbers, but because the circulating EV pool becomes more adept at preventing ischemic damage. In the result section, we have revised this part:
“The concentration of circulating NIC, RIC, BFRRE and HLRE EVs did not change at any timepoint after intervention. Only minor, non-significant fluctuations were observed” (page 5, L179-81)
Furthermore, we have added the following sentence to the discussion:
“Consequently, RIC and BFRRE most likely induced release of EVs with an altered composition of cargo and/or surface markers, and as such, the circulating EV pool became more adept in preventing ischemic damage” (page 13, L471-3)
- Figure 2: The cell photos will definitely need to be larger or reshot with a contrast enhancer. The morphology in panels DEFG is particularly hard to see. Panel C indicates that only RIC is effective at preventing death during OGD insult. Is this correct? Comparisons between NIC and HLRE in panel C must be done…or were they not significant? This isn’t clear from the image itself.
Thank you for pointing this out. We have uniformly adjusted contrast (98), brightness (-13) and RGB levels (15, 0.9, 235) for all 4 pictures to make the tubes more visible. The ImageJ angiogenesis analysis plugin result is inserted in the corner showing the colored dots of branching and the branch length lines.
Furthermore, we agree that panel C could have been made more easily interpretable. In panel C, we show that EVs released immediately (5 min) after RIC protect cells against ischemic damage (increased viability compared to Pre EVs and compared to NIC (5 min)). The same was true for EVs released immediately after BFFRE. EVs released immediately after HLRE (5 min) significantly reduced ischemic damage when compared to the pre EVs from the same samples/subjects. However, this was non-significant when compared to NIC (5 min) (p = 0.15). Thus, as we included a NIC group to make the results more robust, we cannot conclude that HLRE is effective in sustaining cell viability after ischemic damage. We could have annotated all comparisons that were non-significant with “NS”. However, we believe that this would have made the graph cluttered and reduced the immediate interpretability. That is, only comparisons that were significant (p < 0.05) was annotated (page 7).
- L167: Change “subjects” to “samples”
Thank you for pointing out that this does not clear describe the n in the data. We have rephrased it to: “samples from 6 volunteers in each group”. (Page 7, L235+L244)
- L174-188: Angiogenic potential is usually measured molecularly. Treatment with EVs then measurement for VEGF or some other angiogenic factor via Western blotting may be more authoritative than low-contrast morphological images. However, the failure to see changes may preclude the need for this…if you have time, you can try but it isn’t 100% necessary.
This is a valid comment. As the reviewer states, we did not see significant differences in the angiogenic response using the Matrigel assay and therefore did not include additional molecular measurements
- L189-215: After having 3 interventions, the focus suddenly shifts to RIC only. Is there some reason for this? If so, please indicate why only RIC was tested from this point forward.
Of the 3 interventions RIC is the only clinically relevant treatment of acute stroke in the clinic. We therefore chose RIC EVs in the acute studies presented here. Longer term follow-up after acute stroke (months) could warrant the inclusion of exercise EVs in a translational context. We have added the justification to the Results section: “As the RIC intervention is the most clinically relevant for acute ischemic stroke patients, for whom exercise is difficult to perform, we chose to only include RIC EVs in the acute transient MCAO stroke model” (page 8, L282-4)
- Figure 3: Good results, just needs to be bigger so the images can be seen clearly. Also, please clearly label the ischemic hemisphere on the images.
Thank you. We have inserted an “i” on the right hand side of the brains, indicating the insertion of the filament and induction of transient stroke. In addition, the pictures of the brains have been enlarged (page 9)
- Figure 4: Are any of these results significant? If so, please use a commonly accepted method of indicating significance here.
Unfortunately, none of the laser speckle data reached significance when comparing vehicle treated animals to EV treated animals as well as comparisons between pre- and post-RIC EV groups.
- L248-261: The lack of improvement or protection seems to run counter to the hypothesized effect of RIC-induced EVs on stroke. This is why molecular characterization is so important in determining the effect (or lack thereof) in proposed clinical interventions.
We agree with the reviewer that molecular characterization could have been a good complement to the negative results. In this study, we focused on the effect of conditioned EVs in preventing injury in cells and the brain from oxygen deprivation/ischemia. The endpoints are mainly on a macro level. However, we didn’t see a promising effect from our in vivo study with the animal model. Further molecular studies could pursued in the cell assays where we do see a protection after treatment with conditioned EVs.
- Figure 6: The lack of significance here is troubling and not explained due to a lack of molecular modeling. Is there any way to indicate the penumbral area of lower ischemia? Would this make a difference?
Unfortunately, none of our infarct area/volume measurements showed any significant differences between animals treated with vehicle, pre- or post-RIC EVs. Due to the use of anti-Map2 stainings for these measurements, it is not possible to distinguish between penumbra and infarct core.
Discussion
The honesty in the Discussion is appreciated. Too many reports attempt to hide negative data and this hurts the field by forcing other groups to needlessly repeat experiments to arrive at the same conclusion. However, more mechanistic references are needed to provide a picture of why EVs underperformed in this report. Notably, the differences in the cell line and animal experiments provide the main justification for exploring mechanisms; if protein/mRNA expression profiles were similar in the cells and mice, this points to some unknown mechanism but if they are different, then the poor showing of EVs in mice can be further explored by deletion/KO studies in the genes that are different.
- L402-415: Please include what is known about targeting and how EVs chemotaxis occurs (include molecular mechanisms)
Thank you for suggesting the inclusion of this important point. We have added a paragraph to the Discussion: “EVs have been reported to impact physiological and pathological processes, including inhibition of immune response [61], inhibition of tumor growth [62], and stimulation of cell differentiation [63]. The surface protein components and lipid composition of EVs have been found to play a role in targeting and signaling of EVs. In cancer, it has been seen that the composition of integrins on tumor derived EVs could decide the targeting organ of the EVs and initiate pre-metastasis at the target site. In addition, fusion of specific ligands/peptides into EV membrane can achieve efficient EV targeting, such as fusion of neuron-specific RVG peptide leads to EVs targeting the brain, while fusion of ischemic myocardium-targeting peptide makes EVs target ischemic injured cardiomyocytes. In our study, the different behavior of conditioned and non-conditioned EVs suggests that conditioning changes surface proteins on the EVs to facilitate targeting to hypoxic areas.” (page 16, L538 – page 16, L559)
- L441: Erase the extra space before the period here
Thank you for pointing this out. It has now been corrected
- L455: The effect of reactive oxygen species is not mentioned here but is a key concern in reperfusion injury. Additionally, mitochondrial damage and necrosis mechanisms are needed to understand possible intervention points for EVs.
Thank you for point this out. A further discussion with the possible involvement of reactive oxygen species, potential intervention targets of EVs that can regulate mitochondrial damage and necrosis have been added there: “In addition to increased tissue oxygenation, the indication of acute tissue protective effects by post-RIC EVs could also be related with decreased oxidative stress by reducing the production of mitochondrial reactive oxygen species [79] or upregulating the Nrf2-ARE pathway, therefore increase the expression of antioxidant proteins [14]. Besides, reduced inflammatory response by suppressing NFκB activation and TNFα, decreased apoptosis by activation of STAT3 could also contribute to the effects of RIC” (page 16, L604-10)
- L457-464: Mouse models of exercise to exhaustion exist. This reference may help plan future studies that do not carry the mystery of interspecies EV transfer. PMID: 27286034
The reviewer is correct that mouse models on exercise to exhaustion exist. They are however confined to voluntary running, whereas current mouse models of loaded exercise (i.e., resembling resistance exercise) cannot be considered voluntary – or only if based on motivation via feeding. We aim to mimic loading and ischemia-reperfusion of loaded exercise through voluntary high-load and low-load resistance. Therefore, we do not feel that the reference constitutes an ideal reference for future studies. However, we fully admit that our phrasing on this in the discussion was too esoteric and have now elaborated in the discussion (page 16, L616-24)
Methods
- No justification for the timepoints is given. Why jump from 30 min to 6 weeks? Why not 24 hours? Or 18 months?
We thank the reviewer for the comment. We agree that we should have been more explicit on our rationales of choice of time points selected for analysis. We have now elaborated as follows in the manuscript: “Blood sampling were collected during the single bout session before and after intervention. Blood sampling at the 30 min time point was based on previously unpublished pilot time course experiments, indicating that EV were prone to increase and/or infer cell culture biological effects at that time point. Whereas the 30 min time point provided a measure of the acute response to single treatment intervention, the 6 week time point (i.e., upon finalization of the prolonged repeated treatment/training protocol), allowed for a potential readout of chronic changes in the basal level of selected outcomes.” (page 17, L684-90)
- Were these patients controlled for factors (like smoking, which has been shown to change the nature of EVs PMID: 28063900) that could affect EV variability?
This is a very valid point. We have described the exclusion criteria in details: “as well as the exclusion criteria. Briefly, exclusion criteria were; (1) resistance training within 6 months prior to inclusion; (2) participation in moderate/high intensity exercise training (other than resistance training) more than 1 h/week 6 months prior to inclusion; (3) smoking, and; (4) use of prescription medication or intake of dietary supplements potentially affecting muscle metabolism and growth. Moreover, volunteers were instructed to maintain their habitual level of physical activity during the intervention period and to refrain from strenuous physical activity and alcohol for 3 days prior to all tests including the single bout session. The volunteers were asked to fast overnight prior to all test days” (page 17, L659-67)
- Oddly enough, EV utility in ischemia has been reported to be gender-dependent (DOI: /10.1038/s41598-020-69297-0). Would female mice have responded better?
It’s true that including female mice could have added to this study. Female mice might respond differently to the EV treatment. Gender differences have been reported in EV utility as biomarkers for cardiovascular disease (DOI: /10.1038/s41598-020-69297-0) and also as treatment after brain disease (https://doi.org/10.1089/neu.2019.6443). However, in studies that investigates gender differences in remote ischemic conditioning or ischemic conditioning, controversial results were reported. Some reported no gender differences (https://doi.org/10.3389/fphys.2021.667961), while others found males responded better to ischemic conditioning compared to females, which might be due to the confounding effect of sex hormones (https://doi.org/10.1038/s41598-021-03003-6).
- Figure 7 is very helpful
Thank you. To help understand the study, we have referred to the figure at the end of the introduction. In this way, the figure is now figure 1 (page 4)
- L478-479: Please adjust the size of these numbers to match the rest of the text.
Thank you for pointing this out. It has now been corrected
- L538: Please adjust the size of the country name here (and in other places) to the same size as the rest of the text.
Thank you for pointing this out. It has now been corrected
- L652: Having the permit number here (and on L478) is excellent.
Thank you.
Round 2
Reviewer 1 Report
My questions had been well addressed, this submission is acceptable with this current vision.
Author Response
We thank the reviewer for the work of reviewing our paper
Reviewer 2 Report
The authors are to be commended again for making these extensive changes to the manuscript. Only minor issues remain. Even though negative, this paper could serve as an important foundation for the development of animal or other clinical models that can be used to fully explore the concept of pre-conditioning for EV protection during expected cardiovascular surgery.
Minor Corrections:
In Fig. 4, please relabel the brain spheres as the "I" isn't visible on the evaluation PDF and neither is the scale bar.
Recommend proofing by a native speaker.
Author Response
We thank the reviewer for all the work in reviewing our paper. It has been a great addition to the manuscript.
We have inserted a new version of figure 4 with the "i" indication of the ischemic hemisphere and the scalebar visible.